# The Role of MRI and PET/CT in Radiotherapy Target Volume Determination in Gastrointestinal Cancers—Review of the Literature

**DOI:** 10.3390/cancers15112967

**Published:** 2023-05-29

**Authors:** Ajra Secerov Ermenc, Barbara Segedin

**Affiliations:** 1Department of Radiation Oncology, Institute of Oncology Ljubljana, 1000 Ljubljana, Slovenia; bsegedin@onko-i.si; 2Faculty of Medicine, University of Ljubljana, 1000 Ljubljana, Slovenia

**Keywords:** PET/CT, MRI, target volume determination, interobserver variability, anal canal cancer, esophageal cancer, rectal cancer, pancreatic cancer

## Abstract

**Simple Summary:**

Gastrointestinal cancers represent a major burden in oncology worldwide. As radiotherapy is a cornerstone of the treatment strategies, accurate treatment planning is necessary. Imaging modalities, such as positron emission tomography with computed tomography (PET/CT) and magnetic resonance (MRI) could improve target volume determination. This review summarizes the role of PET/CT and MRI in radiotherapy treatment planning for anal, esophageal, rectal and pancreatic cancer.

**Abstract:**

Positron emission tomography with computed tomography (PET/CT) and magnetic resonance imaging (MRI) could improve accuracy in target volume determination for gastrointestinal cancers. A systematic search of the PubMed database was performed, focusing on studies published within the last 20 years. Articles were considered eligible for the review if they included patients with anal canal, esophageal, rectal or pancreatic cancer, as well as PET/CT or MRI for radiotherapy treatment planning, and if they reported interobserver variability or changes in treatment planning volume due to different imaging modalities or correlation between the imaging modality and histopathologic specimen. The search of the literature retrieved 1396 articles. We retrieved six articles from an additional search of the reference lists of related articles. Forty-one studies were included in the final review. PET/CT seems indispensable for target volume determination of pathological lymph nodes in esophageal and anal canal cancer. MRI seems appropriate for the delineation of primary tumors in the pelvis as rectal and anal canal cancer. Delineation of the target volumes for radiotherapy of pancreatic cancer remains challenging, and additional studies are needed.

## 1. Introduction

Gastrointestinal cancers, including colorectal, esophageal, gastric, pancreatic, hepatocellular and biliary tract cancers, are the most common cause of cancer mortality worldwide. The estimated number of new cases of gastrointestinal cancers in 2020 was 5,142,192, and colorectal carcinoma remains the third most common cancer worldwide, with an incidence of 1,931,518 cases in 2020. Pancreatic cancer is still a disease with poor prognosis with an estimated 466,003 death cases worldwide in 2020 [1]. Therefore, gastrointestinal cancers remain an important burden in oncology, and new treatment strategies are warranted. Radiotherapy is one of the cornerstones of cancer treatment besides surgery and systemic therapy. Accurate tumor delineation in radiotherapy treatment planning is crucial to ensure appropriate target coverage and local control of the disease. In recent years, several new technical developments have emerged, such as image-guided radiotherapy (IGRT), intensity-modulated radiotherapy (IMRT), stereotactic body radiotherapy (SBRT) and proton beam therapy (PBT), the so-called “precision radiotherapy” modalities. The delivery of highly conformal and high-dose radiotherapy requires even more accurate imaging and treatment planning. Computed tomography (CT) is the standard imaging modality for radiotherapy treatment planning; however, it presents some limitations, and tumor delineation can be inaccurate, especially for soft tissues. Therefore, other imaging modalities appropriate for radiotherapy treatment planning have emerged, in particular, positron emission tomography (PET) and magnetic resonance imaging (MRI) [2,3,4].

PET is a functional imaging technique involving the use of radioactive tracers—positron emitters attached to targeted biologically active molecules. To date, the most commonly used tracer in tumors of the gastrointestinal tract is 18-F-fluorodeoxyglucose (FDG), an analogue of glucose, which accumulates in tissues with a high metabolic turnover. Currently, FDG-PET/CT is considered an essential diagnostic method for the initial staging of esophageal and anal cancers, and it could also play a role in assessing the response to chemoradiotherapy [5,6]. There is no evidence supporting routine use of FDG-PET/CT in the initial staging of colorectal, gastric and pancreatic cancers, but it could be useful for patients in whom conventional imaging is non-conclusive regarding distant metastases [7].

The role of PET/CT in radiotherapy treatment planning is still under investigation; however, a better understanding of tumor biology with new functional imaging techniques could have an impact on the target delineation, a concept called biological target volume (BTV). BTV takes into account the metabolic, biochemical and physiological changes within a tumor, making it possible to deliver a higher dose of radiation to the BTV, rather than the whole tumor, in a process called dose painting. Delivering higher doses in radioresistant areas, such as a hypoxic region, could result in better tumor control [8,9].

Other PET tracers have been developed, including fluorine-18 fluoromisonidazole (F-MISO), which binds to the hypoxic areas of a tumor, and 3′-deoxy-3′-fluorothymidine (FLT), which is a measure of the tumor proliferation rate [2,9]. Recently, another promising tracer, the fibroblast activation protein inhibitor (FAPI), has been developed. FAPI facilitates the visualization of FAP-expressing cancer-associated fibroblasts [10].

On the other hand, MRI utilizes the magnetic properties of hydrogen protons within a tissue to generate image contrast. It provides higher resolution and greater soft tissue contrast when compared to CT imaging for some tumor sites, for example pelvic tumors. However, MRI can be more challenging for other tumor sites, such as the upper abdomen or mediastinum, due to organ motion artefacts and the central location, which causes a reduced receiver coil sensitivity. Over the past decade, several technical innovations have improved the image artefacts, such as the use of automatic gating navigators or multi-channel receiver coils.

Diffusion-weighted MRI (DWI) has already been widely adopted in cancer imaging due to its ability to enhance soft-tissue contrast [2,11,12].

The purpose of this review is to provide a summary of the evidence and compare the role of PET/CT or MRI in radiotherapy treatment planning for gastrointestinal cancers, mainly for anal, esophageal, rectal and pancreatic cancer, where the role of radiotherapy in the treatment strategy is more prominent [13,14,15,16].

## 2. Methods

This systematic review was performed using structured search terms following the Preferred Reporting Items for Systematic Reviews and Meta-Analysis (PRISMA) guidelines [17].

We searched PubMed databases with the search terms ((esophageal cancer) OR (rectal cancer) OR (anal cancer) OR (pancreatic cancer)) AND (MR OR MRI OR (magnetic resonance) OR (Positron Emission Tomography) OR (PET CT) OR PET) AND ((radiotherapy treatment planning) OR (interobserver variation) OR (interobserver variability) OR (delineation) OR (contouring) OR (target volume determination)). An additional search of the reference lists of the related articles was also performed. We assessed the full text to determine the eligibility of the articles. Studies were considered eligible if they fulfilled the following three criteria: (1) studies including patients with anal, esophageal, rectal or pancreatic cancer, (2) studies including PET/CT or MRI for radiotherapy treatment planning and (3) studies reporting interobserver variability or changes in treatment planning volume due to different imaging modalities or the correlation between the imaging modality and histopathological specimen. All the searches were limited to full-text articles and studies published in English.

## 3. Results

The search was conducted with a start date of the studies on 31 January 2003 to 31 January 2023. An initial search of the literature retrieved 1396 results, 1318 studies were excluded after reviewing the titles and abstracts. Seventy-eight were considered eligible for analysis, but forty-three were eliminated upon full-text review. We retrieved six articles from an additional search of the reference lists of related articles. Finally, the remaining 41 studies were included in the review (Figure 1).

### 3.1. Site-Specific Results

#### 3.1.1. Anal Canal Cancer

FDG-PET/CT for anal canal cancer (AC) has an overall sensitivity of 93% and a specificity of 76% [5]. Moreover, NCCN guidelines recommend that FDG-PET/CT should be considered for treatment planning [18].

Identifying lymph node metastases (LNMs) is crucial for curative radiotherapy treatment planning in AC, especially in the context of adequate dose coverage of the involved nodes. Several studies have investigated the potential impact of FDG-PET/CT in target volume delineation in patients with AC who were candidates for curative radiotherapy. Mahmud et al. observed that treatment plans were modified in 12.5 to 59.3% of patients when FDG-PET/CT was used, which consisted mainly of a prescribed radiotherapy dose or field changes [5]. Two other studies showed similar changes in radiotherapy treatment volumes when FDG-PET/CT was considered for planning [19,20]. More recently, di Carlo et al. analyzed the target volumes contoured using CT, MRI and FDG-PET/CT separately. FDG-PET/CT showed that LNMs were not detected using MRI in 14/37 patients (38%). In these 14 cases, dose escalation was performed on the FDG-PET/CT-positive nodes. According to this study, FDG-PET/CT is particularly important for the detection of inguinal LNMs [21]. We can assume due to the moderate specificity and high sensitivity and due to the major changes in treatment planning when FDG-PET/CT is used that it is indispensable for detecting and contouring LNMs in AC in the context of curative radiotherapy for dose escalation.

On the other hand, the elective coverage of regional lymph node stations in definitive radiotherapy treatment is also important. Omitting the treatment of possible microscopic, involved lymph nodes has been associated with a higher risk of local failure [22]. There are different established contouring guidelines for AC referring to IMRT, and there are some differences in the definition of elective radiation volumes for some anatomical regions, such as inguinal nodes [23,24,25]. Therefore, several studies analyzed the pattern of LNMs according to FDG-PET/CT. Frennered et al. concluded that anal tumors with perianal extension more often have LNMs in the inguinal rather than in other regions. Interestingly, no FDG-PET LNMs were identified in the ischiorectal fossa, favouring the British and Radiation Therapy Oncology Group (RTOG) guidelines rather than the Australasian recommendations regarding the elective clinical target volume (CTV) coverage of the entire ischiorectal fossa. Moreover, no FDG-PET LNMs were found in the inguinal area located posterolateral to deep vessels, an area that is included in all the contouring guidelines for AC [26]. Similar results were found by Dapper et al. [27] and Garda et al. [28], which means that this area could potentially be omitted in contouring the elective CTV. According to these studies, the elective CTV defined in the published guidelines could be reduced or modified in order to achieve better coverage of elective lymph node regions and cause less radiation-induced side effects.

Regarding primary tumor contouring, FDG-PET/CT was less sensitive than MRI in identifying T4 disease. According to the analysis, FDG-PET/CT could not provide the necessary anatomical detail required, and MRI remains the modality of choice for primary GTV contouring. FDG-PET/CT is sensitive in identifying the primary tumor but cannot fully characterize it [20]. Another study investigating the role of MRI in radiotherapy treatment planning for anal and rectal cancers was conducted by Bird et al. The gross tumor volume (GTV) contoured using MRI was smaller compared to the CT volumes (reduced by 13cc). The organs at risk (OAR) such as the bladder, penile bulb and genitalia received statistically significantly lower doses when delineated on MR in comparison with CT, which could translate into a reduction in radiation-induced toxicity for MR-based delineation [29].

Furthermore, studies assessing the interobserver variability in delineation between different imaging modalities are also important. Rusten et al. compared the target volume delineation of anal cancer using FDG-PET and MRI with respect to interobserver and inter-modality variability. It showed that GTV on FDG-PET had a lower interobserver variability in terms of Dice coefficients, i.e., 0.80 for FDG-PET and 0.74 for MRI, but they were not significantly different from each other (*p* = 0.053). They concluded that due to the rather good agreement, either modality can be used for the standard target delineation of anal cancer [30].

To summarize, FDG-PET/CT seems essential for detecting and contouring LNMs in AC in the pathological node coverage setting as well as in determining the elective nodal station irradiation. MRI is more appropriate for primary tumor delineation due to its better identification of T4 tumors (Figure 2 and Table 1).

#### 3.1.2. Esophageal Cancer

FDG-PET/CT is the gold standard in the staging of esophageal cancer (EC) because of its ability to detect metastatic disease, including LNMs, with 66% sensitivity and 96% specificity [31]. Both neoadjuvant and definitive chemoradiotherapy have become well-established standards of care in the treatment of non-metastatic EC [32]. Similarly, as in AC, assessing the pattern of LNMs is essential for determining the target for elective nodal irradiation in the definitive setting. Garcia et al. reviewed FDG-PET scans in EC patients to characterize the location of FDG-avid LNMs. According to the location, tumors were divided into the upper and lower EC. The most common LNMs in the upper EC were in the supraclavicular, retrotracheal and paratracheal nodes. The most common LNMs in the lower EC were in the paraesophageal and the gastrohepatic space. Overall, 55% of paraesophageal LNMs were adjacent to the primary tumor [33]. Similar results were also found in two other articles, which can help to determine the elective nodal stations for CTV definition in definitive chemoradiotherapy for EC [34,35]. Moreover, FDG-PET can help identify LNMs that are located outside the recommended radiation fields and thus decrease the risk of missing the target. In the preoperative setting especially, involved-field irradiation showed promising results [36]. In addition, some studies showed that involved-field irradiation could be feasible in definitive radiotherapy with no difference regarding the overall survival or local control in comparison to elective nodal irradiation in definitive chemoradiotherapy [37,38]. Therefore, FDG-PET involved-field radiotherapy could be the ideal compromise between smaller treatment volumes with less toxicity and without increasing the risk of undertreatment.

Further studies analzsed the role of FDG-PET/CT in treatment planning for EC in comparison to CT regarding potential geographic misses. Jimenez et al. compared the target volumes and tumor lengths defined by fused FDG-PET/CT in comparison to CT simulation. The GTVnode was significantly greater on FDG-PET/CT. The Dice similarity coefficient analysis showed excellent agreement for GTVtumor, i.e., 0.72, but it was very low for GTVnode, i.e., 0.25. The study showed that CT simulation, without taking into account FDG-PET/CT information, could leave LNMs outside the radiotherapy treatment volume. Only 55.2% of patients had LMNs evident in both image data sets [39]. Moreover, FDG-PET/CT in the diagnostic setting and radiotherapy planning could affect survival. Metzger et al. conducted a retrospective analysis assessing the survival data of 145 patients with EC. FDG-PET/CT information was included in radiotherapy planning. Univariate analysis showed the use of FDG-PET/CT to be associated with significantly longer local recurrence-free survival (*p* = 0.006), which was also confirmed using a multivariate analysis. The authors concluded that the use of FDG-PET/CT improved patients’ outcomes probably due to the more accurate staging and adequate coverage of the target volumes [40].

On the other hand, Muijs et al. found no local recurrence following CT-based radiotherapy that could have been prevented by FDG-PET/CT. Ninety patients were planned for radiotherapy on the basis of CT simulation; all the patients also had FDG-PET/CT scans prior to radiotherapy. After treatment, the treatment volumes were adjusted based on the FDG-PET/CT when necessary. In 32 patients (36%), >5% of the PET CT-based GTV would have been missed if the treatment planning was CT-based. Local recurrences were seen in 10 patients (11%). There were three in-field recurrences, four regional recurrences outside both CT-based and FDG-PET/CT-based CTV and three recurrences at the anastomosis; none of these recurrences were considered preventable by FDG-PET/CT [41].

Another potentially important role for FDG-PET/CT in EC radiotherapy treatment planning is the determination of the primary tumor. Studies comparing the length of the tumor on preoperative FDG-PET/CT scans and on histopathologic specimens after surgery showed a good correlation [42,43,44]. Furthermore, several studies investigating interobserver variability in the contouring GTV of primary tumors were conducted. Vesprini et al. found a small but significant improvement in interobserver variability when FDG-PET was added to CT-based planning for the identification of the primary tumor GTV in patients with gastro-esophageal carcinoma, and the overlap of contours was 72.7% vs. 69.1% (*p* = 0.05), respectively [45]. On the other hand, some other studies did not confirm the effect of FDG-PET on the improvement in interobserver variability. Schreus et al. observed no differences in the concordance indexes, and the mean concordance indexes for CT-based CTV were 72% vs. 72% for FDG-PET/CT-based CTV. Combining FDG-PET and CT may improve target volume definition with fewer geographical misses as FDG-PET/CT modified tumor delineation in 17/28 subjects (61%) in the cranial and/or caudal direction compared to CT alone [46]. Nowee et al. assessed the delineation variability in the GTV between CT and combined FDG-PET/CT in EC patients in a multi-institutional study. No difference in the generalized conformity index (CIgen) was observed (average of 0.67 on CT and 0.69 on PET-CT). FDG-PET significantly influenced the delineated volume in four out of six cases. According to these data, it seems reasonable that an increased uptake should only be used for tumor localization and not to define precise boundaries [47].

However, in the case of cervical EC, Toya et al. observed a reduced interobserver variability for the delineation of GTV on FDG-PET/CT scans compared with CT images alone. The mean interobserver conformity index (CI) of the GTV_CT_ and GTV_PET/CT_ were 0.39 ± 0.15 and 0.58 ± 0.10, respectively (*p* = 0.005) [48]. Moreover, recently, Li et al. investigated inter- and intraobserver delineation variability in the GTVs of EC based on planning CT with different combinations of multimodal diagnostic images from endoscopy, endoscopic ultrasound (EUS), esophagography and FDG-PET/CT. The intraobserver CIgen among different observers and the interobserver CIgen among different combinations of multimodal images showed significant differences (*p* < 0.001). The intraobserver CIgen for senior radiation oncologists was larger than that for the junior radiation oncologists (*p* < 0.001). The use of multimodal imaging, including CT, endoscopy/EUS, esophagography and FDG-PET/CT for target delineation reduced the interobserver variability [49].

A study conducted by Shi et al. showed that it is also feasible to delineate the GTV of primary thoracic EC with reference to the diagnostic FDG-PET/CT image in the absence of planning FDG-PET/CT images. The GTV_3D_ was contoured on a three-dimensional (3D) CT image without referencing the FDG-PET/CT image. The GTV_PET-ref_ was contoured on the 3D-CT image while referencing the FDG-PET/CT image. The GTV_PET-reg_ was contoured on images derived with the deformable registration of 3D-CT and FDG-PET/CT. No significant difference was found between the GTVs delineated based on visual referencing or deformable registration [50].

MRI is not yet an established method for radiotherapy treatment planning for EC, but it is promising because of its excellent soft tissue contrast; additionally, it is a non-invasive imaging modality. Hou et al. compared the GTV longitudinal length measured using different imaging modalities—CT, T2-weighted MRI (T2 MRI) and DWI—with the pathological lesion length to determine the most accurate imaging modality. They concluded that DWI correlated with pathological lesion lengths more precisely compared to CT or T2 MRI. DWI scans fused with CT images can be used to improve the accuracy of GTV delineation in EC [51].

Wollenbrock et al. evaluated the feasibility of target delineation on T2 MRI and T2 MRI combined with DWI (T2 MRI+DWI) compared with FDG-PET/CT. No differences were observed in CIgen (FDG-PET/CT, 0.68; T2 MRI, 0.66; T2 MRI+DWI, 0.68). The most variation was seen at the cranial–caudal borders, and the addition of DWI to T2 MRI can reduce the variation in the caudal border delineation in gastro-esophageal junction tumors. The study showed that MRI-based GTV delineation of the EC is feasible, with an interobserver variability comparable to that in FDG-PET/CT; a potential pitfall could be a lack of experience in delineation using MRI [52].

Wang et al. investigated the assessment of the primary tumor and regional lymph nodes using FDG-PET/MRI, FDG-PET/CT, MRI and contrast-enhanced CT (CECT). The pathology specimen was used as a reference standard to assess the accuracy of all the imaging modalities. For primary tumor staging, the accuracy of PET/MRI, MRI and CECT in comparison to the pathological specimen was 85.7%, 77.1% and 51.4%, respectively. For lymph node assessment, the accuracy of PET/MRI, PET/CT, MRI and CECT was 96.2%, 92.0%, 86.8% and 86.3%, respectively. 18F-FDG PET/MRI has advantages over 18F-FDG PET/CT, MRI and CECT in the preoperative assessment of primary tumors and regional lymph nodes of EC [53].

FDG-PET/CT seems superior, especially in the detection of LNM in EC; consequently, it is essential when determining CTV as in the elective or involved-field irradiation. The role in determining the primary tumor is not so clear; apparently, FDG-PET/CT is suitable for localizing the tumor. MRI could play a role in GTV determination, but there are insufficient data to date (Table 2 and Figure 3).

#### 3.1.3. Rectal Cancer

Surgery is the cornerstone treatment for rectal cancer (RC), but organ-preserving strategies (watchful waiting) are gaining importance. Total neoadjuvant therapy (TNT) is a reference treatment in locally advanced RC with high-risk criteria, including extramural vascular invasion or the involvement of the mesorectal fascia, but it is also used in the treatment of other tumors with high risk criteria such as T4 tumors or positive lymph nodes on imaging [54]. Accurate delineation is necessary to improve local control and minimize toxicity. CT planning is still the gold standard in rectal cancer radiotherapy, but as MRI has superior soft tissue contrast, it is more suitable for boost strategies in the pelvis. FDG-PET/CT could also have a role in target volume determination for RC. Automatically generated FDG-PET/CT contours showed the best correlation with the surgical specimen in comparison with manual FDG-PET, MRI and CT contours [55].

Several papers explored the role of MRI in treatment planning for RC. One of the first papers by O’Neill et al. reviewed imaging and planning data for patients with locally advanced low RC. Eligible patients were defined using MRI as having cT3 tumors deemed to require abdominoperineal excision or cT4 and cT3 tumors with involved mesorectal fascia. Tumor volumes and location were compared using sagittal pre-treatment MRI and planning CT. MR-based rectal tumor volumes were smaller and thus lay further from the anal verge, facilitating the relative sparing of the anal sphincter [56]. Tan et al. included patients with cT3N0M0 tumors, and they compared the volumes of GTV contoured using CT versus MRI. They observed that tumors in the anal region were not identified on the CT data set due to poor soft tissue contrast and a lack of a tumor mass effect. Moreover, in the region of the recto-sigmoid junction, the tumors were underestimated by 50% due to suboptimal tumor visualization. Tumor invasion into the sigmoid colon was demonstrated only using MRI. In the mid-rectal region, the ratio between the volume of GTV contoured using CT and separately using MRI was approximately 1, indicating good correlation. MRI was important and useful, especially when suboptimal tumor visualization occurred on CT in the sigmoid and anorectal subregions, so they concluded that MRI could avoid geographic misses in rectal cancer delineation [57]. Both studies suggested that MRI improves tumor delineation and reduces the tumor volume compared to CT in rectal cancer. It appears that MRI has an important role for locally advanced RC, especially for lower tumors. For patients who are not willing to undergo abdominoperineal excision and have a permanent stoma, omitting surgery could be an option in the case of a complete clinical response—the so-called watch-and-wait strategy. In this case, treatment planning should also be performed with an MRI from the perspective of sphincter sparing.

The role of FDG-PET/CT in treatment planning for RC has also been analyzed. In a study, they included patients with cT2-4N0-2M0 tumors as well as less advanced stages in comparison to the previously mentioned papers. The GTV volumes based on FDG-PET were significantly smaller [58], which was similar to the study conducted by Brændengen et al. [59]. Buijsen et al. showed that the influence of FDG-PET was no different between low- and high-seated tumors and helped to enable tailoring treatment fields, especially in the cranio-caudal direction. As the study included tumors that were also in less advanced stages and FDG-PET helped to determine the cranio-caudal borders, we can assume that FDG-PET could be adequate in treatment planning not only for large tumors but also for less advanced ones. In this clinical scenario, complete response could be achieved with different radiotherapy treatment regimens (TNT, neoadjuvant chemoradiotherapy or short-course radiotherapy), and surgery could be avoided. FDG-PET/CT seems feasible for determining the GTV in smaller tumors for watch-and-wait protocols.

The same study showed that in up to 29% of patients, the CTV based on FDG-PET extended outside the CTV used in clinical practice [58]. However, there have been no other relevant studies assessing the role of different imaging modalities for the delineation of LNMs in RC, as dose escalation for LNMs is not widely accepted in the treatment protocols.

Several studies investigated interobserver variability for different imaging modalities in order to achieve better agreement in delineation for RC between radiation oncologists. Two studies compared the GTV delineated on T2 MRI and DWI images to assess whether the agreement was improved by DWI. The results showed a smaller target volume for DWI, which did not translate into a better agreement [60,61]. Burbach et al. further analyzed the contouring GTV of rectal cancer on MRI sequences. Three observers independently delineated GTV on T2 MRI, DWI and on a combination of both. They observed no differences in the conformity indexes per modality. Delineation on DWI resulted in the smallest volume. To eliminate the geometrical distortion of DWI, Burbach et al. used a special sequence that allowed the direct registration of images to anatomical MRI images, so contouring was more feasible. DWI seemed to have great potential for tumor determination in RC, but more experience in the delineation and elimination of geometrical distortions is needed [61]. Recently, Hearn et al. also assessed the interobserver variability for T2 MRI and DWI contours in locally advanced RC. The observers delineated GTV in three different sessions for each patient on T2 MRI imaging only, on DWI only and on co-registered T2 MRI and DWI. Additionally, in the registered session, they delineated a smaller sub-volume corresponding to their visual assessment of the greatest diffusion restriction. Furthermore, they evaluated the feasibility of semi-automated DWI sub-volume delineation. The contours of the co-registered session demonstrated significantly lower interobserver agreement than T2 MRI and DWI contours for overlap metrics, while there were no significant differences between T2 MRI and DWI contours. The semi-automated delineation demonstrated moderate agreement with manual consensus delineations of the area of greatest diffusion restriction. The authors concluded that delineation based on semi-automated DWI can standardize sub-volume delineation if the registration between acquisitions is sufficiently accurate. This approach could be applied to dose escalation or dose painting protocols to improve delineation reproducibility [62].

FDG-PET/CT was also assessed in studies analyzing interobserver variability for the delineation process in RC, where the GTV was contoured using CT and FDG-PET/CT sequences. Of the 30 evaluable preoperative patients, the mean GTV_PET/CT_ was significantly smaller than the mean GTV_CT_. FDG-PET/CT significantly increased the interobserver concordance index in contouring GTV compared with CT-only-based contouring: 0.56 versus 0.38 (*p* < 0.001). The study suggests that PET CT could reduce interobserver variability when contouring boost volumes [63]. Similarly, Buijsen et al. analyzed the effect of the use of FDG-PET/CT on the interobserver variability in GTV definition. The conformity indexes increased significantly using FDG-PET, and the best interobserver agreement was observed using FDG-PET auto-contours [58]. Recently, Rosa et al. investigated the potential benefit of functional imaging in the form of DWI and FDG-PET/CT for treatment intensification strategies in RC. Radiation oncologists prospectively delineated GTVs using CT, T2 MRI, DWI and FDG-PET/CT. The mean Dice index was 0.85 for GTV-CT, 0.84 for GTV-T2 MRI, 0.82 for GTV-DWI and 0.89 for GTV-PET (*p* = 0.009). DWI resulted in a smaller volume of delineation compared to the other image modalities. They concluded that DWI could be an optimal strategy for boost volume delineation for dose escalation in patients with RC [64]. According to the presented studies, the use of different MRI sequences did not increase the interobserver agreement, while FDG-PET/CT did.

To sum up, the presented studies showed that MRI could have a role especially in determining the tumor boundaries in the anorectal and sigmoid region. DWI is gaining importance and could have a role in sub-volume treatment planning strategies as it represents the area of greatest diffusion restriction, which could be potentially radioresistant. The addition of FDG-PET/CT could add other important information to the final target volume, could help in tailoring the cranio-caudal border of the tumor and could reduce interobserver variability. Its role in CTV delineation remains investigational (Table 3).

#### 3.1.4. Pancreatic Cancer

The role of radiotherapy in pancreatic cancer (PC) is still controversial. There is a lack of robust evidence, but some patients may still benefit from local treatment. Radiotherapy can be delivered in the neoadjuvant setting for borderline resectable or up-front unresectable tumors. The optimal imaging modality to accurately define the GTV for radiotherapy of pancreatic cancer is unknown. There were only scarce data retrieved from our analysis. Dalah et al. explored the potential role of using various imaging modalities including MR, FDG-PET/CT and CT to define the treatment targets for radiation therapy of PC. Patients were classified into three categories according to whether the tumor was surgically resectable, borderline resectable or locally advanced. Significant differences were found between the volumes of GTV contoured using several modalities. The authors observed the ability of metabolic imaging of FDG-PET to accurately define the GTV. They discussed the accuracy of FDG-PET, which is worse in assessing tumors larger than 4 cm, which is partially due to the low metabolic rates in portions of larger tumors. However, for hypermetabolic tumors smaller than 2 cm, the sensitivity of FDG-PET is superior to that achieved with CT, meaning that FDG-PET could be appropriate for treatment planning for resectable or borderline tumors. The OAR volumes based on MRI are generally smaller than those based on CT. They suggested that further imaging studies with pathological correlation are required to establish an optimal imaging modality for PC [65].

Li et al. compared the differences between CE FDG-PET/CT and CECT in target volume delineation in the treatment plan for locally advanced pancreatic cancer (LAPC). Twenty-one consecutive patients with LAPC underwent both non-CECT and FDG-PET scans; eleven of them also underwent CECT scans. The GTVs were smaller when contoured on FDG-PET/CT scans in comparison to CECT or non-CECT. The co-registration of FDG-PET/CT with CECT could improve the accuracy of GTV delineation in LAPC and could reduce the adverse effect of irradiation [66].

In another study, the role of FDG-PET/CT in the delineation process for unresectable locally advanced PC (LAPC) in comparison to CT was also analyzed. Changes in GTV delineation were necessary in five patients based on FDG-PET/CT information. In these patients, the average increase in GTV was 29.7% due to the inclusion of additional LNMs and extension of the primary tumor beyond that defined by CT. Therefore, the co-registration of FDG-PET and CT images in unresectable LAPC can improve the delineation of GTV and theoretically reduce the likelihood of geographical misses [67].

Liermann at al. investigated a novel tracer, FAPI, in tumor volume determination in LAPC. The GTVs of seven patients with LAPC were contoured by six radiation oncologists. Additionally, FAPI-PET/CT was used to automatically delineate the GTV. There was no significant difference between the volumes of automatic FAPI-GTVs and most of the GTVs manually contoured by radiation oncologists. They concluded that FAPI-PET/CT can be used as an additional imaging modality to improve decision-making in target definition [68].

Functional imaging with FDG-PET or other tracers could have a role in target volume determination for LAPC, but further investigations are warranted (Table 4).

## 4. Conclusions and Future Directions

Accurate tumor delineation is necessary as precision radiotherapy is becoming important in the treatment strategies for gastrointestinal cancers, especially anal, esophageal, rectal and pancreatic cancer.

FDG-PET/CT seems especially essential for LNM delineation and the definition of the elective nodal irradiation in AC and EC due to its accuracy. The addition of FDG-PET/CT modifies the CTV and could potentially avoid a geographic miss and therefore allow for smaller treatment fields without risking undertreatment. FDG-PET/CT helps to define the regional lymph node stations at greater risk of micrometastasis and thus the elective treatment volume in definitive chemoradiotherapy. Its role in primary tumor delineation for EC is not as clear. It seems that FDG-PET/CT is appropriate for tumor localization but not for defining precise boundaries, for which another imaging modality could be more adequate, for instance MRI, but there is not sufficient evidence.

To date, it seems that MRI has an important role in determining the primary tumor, especially in the pelvis, for the delineation of AC and RC. MRI for GTV tumor delineation in AC could reduce the target volume and consequently reduce the dose to OAR. Similarly, MRI has an important role in tumor target delineation in RC, where it can help to define the GTV in the anorectal and sigmoid region. FDG-PET/CT for GTV determination in RC could tailor the craniocaudal border delineation and improve interobserver agreement.

It appears that especially DWI sequences are becoming increasingly important in boost treatment planning strategies. DWI enables GTV definition with a high level of precision and allows dose escalation to the area of the greatest restriction of diffusion.

Interestingly, in our search, we retrieved fewer articles regarding PET/CT and MRI for target volume determination in RC in comparison to EC. One of the reasons could be that the reference treatment for locally advanced RC is still neoadjuvant therapy (TNT, chemoradiotherapy or short-course radiotherapy) followed by surgery, so the focus of the studies is on the optimal treatment strategy rather than imaging for target volume delineation. In addition, MRI is the gold standard for diagnostic imaging in RC, and probably the scientific community does not feel the need to perform studies comparing MR and CT in this setting. Moreover, dose escalation for LNM in RC is not a routinely adopted strategy in standard treatment protocols. However, as organ sparing in RC is gaining importance, dose escalation to LNMs could be a future strategy, and new studies are needed.

Target definition in PC is challenging due to the anatomical and physiological specificity of the region—sometimes it is difficult to distinguish the tumor from fibrosis in PC. MRI and FDG-PET/CT could improve target volume delineation in PC, but to date, there is insufficient data. The use of other tracers is still investigational.

In conclusion, both PET/CT and MRI are gaining an important role in radiotherapy target volume determination for gastrointestinal cancers, for the delineation of the primary tumor, pathological lymph nodes and elective volume definition. Further studies to determine the optimal combination of imaging modalities for radiotherapy treatment planning are warranted. Whenever possible, studies with pathohistological confirmation should be performed, as they represent the real correlation between the tumor and different imaging modalities. Research on interobserver variability with different imaging modalities is also recommended, especially with a dosimetric analysis.

## Figures and Tables

**Figure 1 cancers-15-02967-f001:**
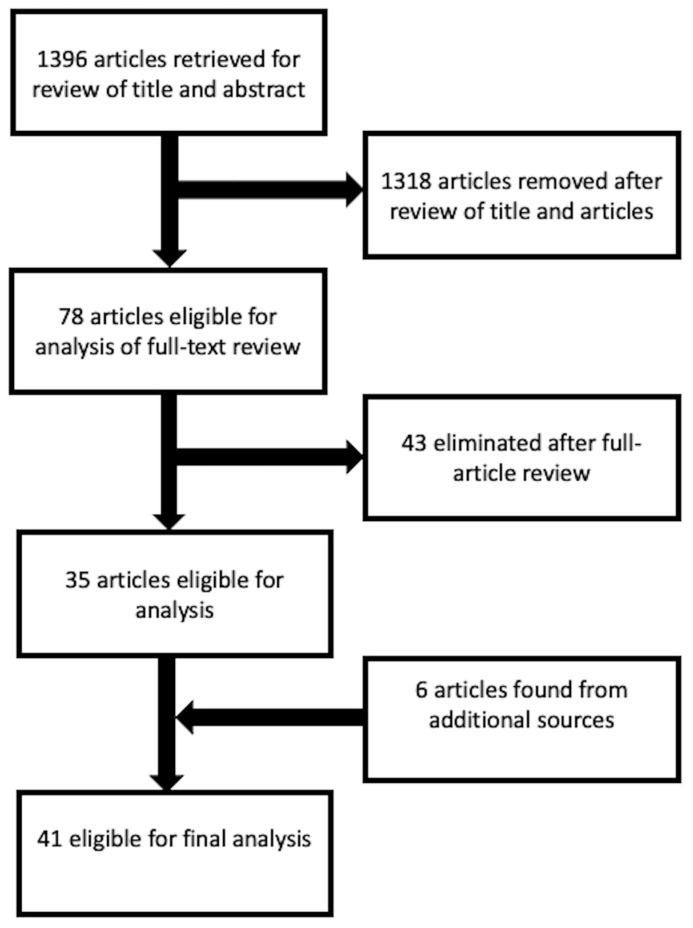
Search flow-chart according to the PRISMA guidelines.

**Figure 2 cancers-15-02967-f002:**
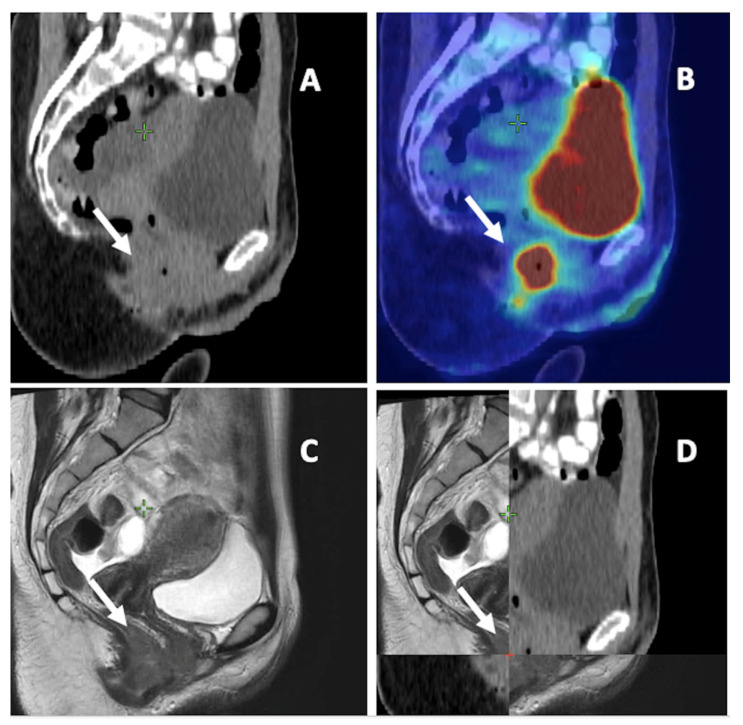
Anal cancer on different imaging modalities. (**A**) Computed tomography (CT). (**B**) 18-F-fluorodeoxyglucose positron emission tomography CT. (**C**) Magnetic resonance imaging (MRI). (**D**) a combination of CT and MRI. White arrow: primary tumor of anal canal.

**Figure 3 cancers-15-02967-f003:**
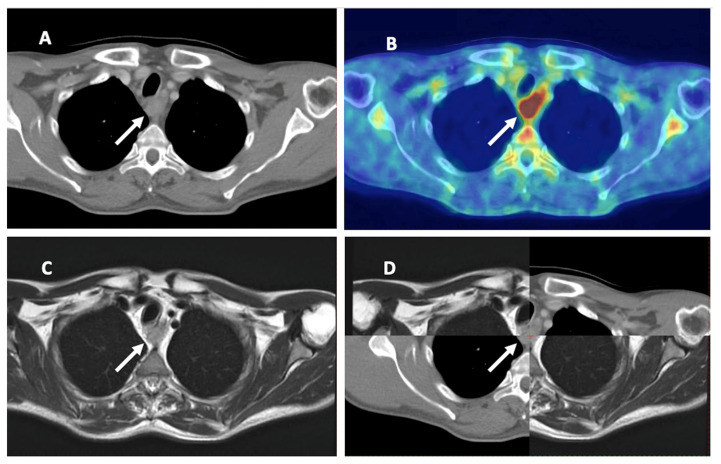
Esophageal cancer on different imaging modalities. (**A**) Computed tomography (CT). (**B**) 18-F-fluorodeoxyglucose positron emission tomography CT (FDG-PET/CT). (**C**) Magnetic resonance imaging (MRI). (**D**) combination of CT and MRI. White arrow: primary tumor of the esophagus.

**Table 1 cancers-15-02967-t001:** 18-F-fluorodeoxyglucose positron emission tomography with computed tomography and magnetic resonance imaging for target volume determination in anal canal cancer.

Study	Design	Patients	Imaging Modality	Observations
Mahmud et al. [5]	meta-analysis	17 studies	FDG-PET/CT, CT	FDG-PET/CT modified treatment plans in 12.5 to 59.3% of pts.
Kregli et al. [19]	prospective	27	FDG-PET/CT, CT	FDG-PET/CT changed GTV in 55.6% and CTV 37.0%.
di Carlo et al. [21]	retrospective	37	FDG-PET/CT, MRI, CT	FDG-PET/CT showed LNMs were not detected on MRI in 38% patients.
Frennered et al. [26]	retrospective	190	FDG-PET/CT, CT	No FDG-PET LNMs were identified in the ischiorectal fossa or inguinal area located posterolateral to the deep vessels.
Dapper et al. [27]	retrospective	37	FDG-PET/CT, CT	Of 49 FDG-PET-positive inguinal nodes, 10–29% were situated outside the recommended guidelines.
Garda et al. [28]	retrospective	40 (79 inguinal nodes)	FDG-PET/CT, MRI, CT	No LNMs were identified lateral or posterior to the vessels in the inguinal region.
Zimmermann et al. [20]	retrospective	26	FDG-PET/CT, MRI	FDG-PET/CT led to major changes in treatment planning in 17% pts, and MRI was more sensitive in identifying T4 disease.
Bird et al. [29]	prospective	17	MRI, CT	GTV and PTV smaller on MRI compared to CT, and dose to OAR was significantly lower on MRI in comparison to CT.
Rusten et al. [30]	prospective	19	FDG-PET/CT, MRI	Dice coefficients of 0.80 for FDG-PET and 0.74 for MRI, (*p* = 0.053).

Abreviations: FDG-PET/CT = 18-F-fluorodeoxyglucose positron emission tomography with computed tomography; MRI = magnetic resonance imaging; GTV = gross tumor volume; LNM = lymph node metastasis; CTV = clinical target volume; PTV = planning target volume; CT = computed tomography; OAR = organ at risk; pts = patients.

**Table 2 cancers-15-02967-t002:** 18-F-fluorodeoxyglucose positron emission tomography with computed tomography and magnetic resonance imaging for target volume determination in esophageal cancer.

Study	Design	Patients	Imaging Modality	Observations
Garcia et al. [33]	retrospective	473	FDG-PET/CT	Most common LNMs in upper EC—supraclavicular, retrotracheal and paratracheal. Most common LNMs in lower EC—paraesophageal and in gastrohepatic space. There were 55% paraesophageal LNMs adjacent to the primary tumor.
Machiels et al. [34]	retrospective	105	FDG-PET/CT	Good correlation between distribution of nodal volumes at risk in surgical series and on FDG-PET/CT.
Münch et al. [35]	retrospective	76	FDG-PET/CT	Most common sites of LNMs—paraesophageal and paratracheal, and <5% of patients had supraclavicular, subaortic, diaphragmatic or hilar LNMs.
Jimenez et al. [39]	retrospective	29	FDG-PET/CT, CT	Dice similarity coefficient of 0.72 for GTVtumor and 0.25 for GTVnode.
Metzger et al. [40]	retrospective	145	FDG-PET/CT	FDG-PET/CT included into radiotherapy planning was associated with significantly longer local recurrence-free survival.
Muijs et al. [41]	prospective	90	FDG-PET/CT, CT	Local recurrences were seen in 10 patients (11%); none were considered preventable by FDG-PET/CT.
Han et al. [42]	prospective	22	FDG-PET/CT	SUV cut-off of 2.5 on FDG-PET/CT provided closest estimation of GTV length.
Mamede et al. [43]	retrospective	34	FDG-PET/CT	FDG-PET-derived tumor length of untreated EC correlated well with surgical pathology results.
Zhong et al. [44]	prospective	36	FDG-PET/CT	SUV cut-off of 2.5 provided closest estimation of tumor length.
Vesprini et al. [45]	prospective	10	FDG-PET/CT, CT	Overlap of contours was 72.7% for FDG-PET/CT vs. 69.1% for CT alone (*p* = 0.05).
Schreus et al. [46]	retrospective	28	FDG-PET/CT, CT	Mean concordance indexes for CT-based CTV and FDG-PET/CT-based CTV were 72%.
Nowee et al. [47]	retrospective	6	FDG-PET/CT, CT	No difference in CIgen was observed (average 0.67 on CT, 0.69 on PET-CT).
Toya et al. [48]	retrospective	10	FDG-PET/CT, CT	Mean interobserver CI of GTV_CT_ and GTV_PET/CT_ was 0.39 ± 0.15 and 0.58 ± 0.10 (*p* = 0.005), respectively.
Li et al. [49]	prospective	51	FDG-PET/CT, CT, EUS, endoscopy, esophagography	Multimodal imaging (CT, endoscopy/EUS, esophagography, FDG-PET/CT) reduced interobserver variability.
Shi et al. [50]	prospective	72	FDG-PET/CT, CT	No significant difference between the GTVs delineated based on visual referencing or deformable registration.
Hou et al. [51]	prospective	42	MRI, CT	DWI displayed EC lengths most precisely when compared with CT or regular MRI.
Vollenbrock et al. [52]	prospective	6	FDG-PET/CT, MRI	No differences were observed in CIgen (FDG-PET/CT, 0.68; T2 MRI, 0.66; T2 MRI+DWI, 0.68).
Wang et al. [53]	prospective	35	FDG-PET/CT, CECT, MRI	For primary tumor staging, accuracy of PET/MRI, MRI and CECT in comparison to the pathological specimen was 85.7%, 77.1% and 51.4%.

Abbreviations: FDG-PET/CT = 18-F-fluorodeoxyglucose positron emission tomography with computed tomography; MRI = magnetic resonance imaging; GTV = gross tumor volume; LNM = lymph node metastasis; CTV = clinical target volume; SUV = standardized uptake value; CT = computed tomography; CIgen = generalised conformity index; CI = conformality index; EUS = endoscopic ultrasound; DWI = diffusion-weighted MRI; EC = esophageal cancer; T2 MRI = T2-weighted MRI; CECT = contrast-enhanced CT.

**Table 3 cancers-15-02967-t003:** 18-F-fluorodeoxyglucose positron emission tomography with computed tomography and magnetic resonance imaging for target volume determination in rectal cancer.

Study	Design	Patients	Imaging Modality	Observations
O’Neill et al. [56]	retrospective	10	MRI, CT	Tumor volumes defined on MRI were smaller and more distant from the anal sphincter than CT-based volumes.
Tan et al. [57]	retrospective	15	MRI, CT	MRI was useful where suboptimal tumor visualisation occurred on CT—in sigmoid and anorectal subregion.
Regini et al. [60]	retrospective	27	MRI	Results showed smaller target volume on DWI, which did not translate into better agreement.
Burbach et al. [61]	prospective	24	MRI	No differences in CI were observed per modality (T2 MRI and DWI). Smallest volume was delineated using DWI.
Hearn et al. [62]	retrospective	20	MRI	Contours of co-registered session (T2 MRI and DWI) demonstrated significantly lower interobserver agreement.
Buijsen et al. [55]	prospective	26	FDG-PET/CT, MRI, CT	Automatically generated FDG-PET/CT contours showed best correlation with surgical specimen compared to manual FDG-PET, MRI and CT contours.
Whaley et al. [63]	retrospective	34	FDG-PET/CT, CT	FDG-PET/CT increased CI in contouring GTV compared with CT only: 0.56 versus 0.38 (*p* < 0.001).
Buijsen et al. [58]	retrospective	42	FDG-PET/CT, MRI, CT	CI increased significantly using PET, best interobserver agreement was observed using PET auto-contours.
Brændengen et al. [59]	prospective	68	FDG-PET/CT, MRI	Median volume of GTV-MRI was larger than GTV-PET (*p* < 0.001).
Rosa et al. [64]	retrospective	27	FDG-PET/CT, MRI, CT	Mean Dice index was 0.85 for GTV-CT, 0.84 for GTV-T2 MRI, 0.82 for GTV-DWI and 0.89 for GTV-PET (*p* = 0.009). DWI resulted in smaller volume.

Abbreviations: FDG-PET/CT = 18-F-fluorodeoxyglucose positron emission tomography with computed tomography; MRI = magnetic resonance imaging; GTV = gross tumor volume; LNM = lymph node metastasis; CT = computed tomography; DWI = diffusion-weighted MRI; T2 MRI = T2-weighted MRI; CI = conformity/concordance index.

**Table 4 cancers-15-02967-t004:** Positron emission tomography with computed tomography and magnetic resonance imaging for target volume determination in locally advanced pancreatic carcinoma.

Study	Design	Patients	Imaging Modality	Observations
Dalah et al. [65]	retrospective	19	FDG-PET/CT, MRI	Significant differences were found between volumes of GTV contoured on several modalities.
Li et al. [66]	prospective	21	FDG-PET/CT, CECT	GTVs were smaller when contoured on FDG-PET/CT scans in comparison to CECT or non-CECT.
Topkan et al. [67]	prospective	14	FDG-PET/CT, CT	Changes in GTV delineation were necessary in five patients based on FDG-PET/CT information. Average increase in GTV was 29.7%.
Liermann et al. [68]	retrospective	7	FAPI-PET/CT	No significant difference between volumes of automatic FAPI-GTVs and most of manually contoured GTVs.

Abbreviations: FDG-PET/CT = 18-F-fluorodeoxyglucose positron emission tomography with computed tomography; MRI = magnetic resonance imaging; GTV = gross tumor volume; LNM = lymph node metastasis; CT = computed tomography; CECT = contrast enhanced CT; FAPI-PET/CT = inhibitor of fibroblast activation protein PET/CT.

## Data Availability

Data sharing not applicable.

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
