# Peer review of "The Role of MRI and PET/CT in Radiotherapy Target Volume Determination in Gastrointestinal Cancers—Review of the Literature"

_cancers, 2023, doi:10.3390/cancers15112967_

Round 1

Reviewer 1 Report

Updated review and highlighting the need for more controlled and histologically correlated studies to create evidence.

Author Response

Dear reviewer,

We thank you for your time and effort in reviewing our manuscript. The feedback has been important in improving the content of the paper. We have made some major changes in the relevant sections, we have largely extended our discussion and put the articles in a more clinical perspective. We have additionally stressed the importance of the studies investigating the histopathological correlation with different imaging modalities. All authors have read and approved the revised manuscript. We hope that our resubmission is now suitable for inclusion in Cancers and we look forward to hearing from you,

Ajra Secerov Ermenc and Barbara Segedin

Reviewer 2 Report

The authors presented a literature review on the use of PET/CT and MRI in the diagnosis and treatment planning of patients with anal, esophageal, rectal and recurrent pancreatic cancer. A qualitative analysis of the literature was carried out, especially on the evaluation of lymph node metastases, and modern data were presented. The article is of great interest to specialists.

Good day! I have no additional questions for the authors

The article is written in good English, the text is clear, does not cause difficulties in reading.

Author Response

Dear reviewer, 

We thank you for your time and effort in reviewing our manuscript. The feedback has been important to us. We have made some corrections in the contet of the manuscript. Best regards,

Ajra Secerov Ermenc and Barbara Segedin

Reviewer 3 Report

1- Up to date references are needed.

2- This review need to a discussion section.

3- English grammar and spelling need to improvement.

English grammar and spelling need to improvement.

Author Response

Dear reviewer,

We thank you for your time and effort in reviewing our manuscript. The feedback has been important in improving the content of our paper. We have made an additional search in the database with our selected criteria, but we didn't find any more relevant and recent articles. However, we have added several new references because we have largely extended our discussion in the Relevant sections and Conclusions paragraphs. We didn't add an additional paragraph Discussion, as it is not mandatory in the instructions for author's (see attachment), but we have changed the content to a great extent. We have put the discussion in a more critical perspective. Thank you for pointing out the need for grammar and spelling corrections, the manuscript underwent for English revision by a professional. All authors have read and approved the revised manuscript. We look forward to hearing from you, 

Ajra Secerov Ermenc
